# Comorbidity and outcomes in traumatic brain injury: protocol for a systematic review on functional status and risk of death

Tatyana Mollayeva,[1,2,3] Chen Xiong,[1,2,3] Sara Hanafy,[2,3] Vincy Chan,[2,3] Zheng Jing Hu,[4] Mitchell Sutton,[4] Michael Escobar,[4] Angela Colantonio[1,2,3,4]

[1]Rehabilitation Sciences Institute, Faculty of Medicine, University of Toronto, Toronto, Ontario, Canada
[2]Research Department, Toronto Rehabilitation Institute-University Health Network, Toronto, Ontario, Canada
[3]Aquired Brain Injury Research Lab, University of Toronto, Toronto, Canada
[4]Dalla Lana School of Public Health, University of Toronto, Toronto, Ontario, Canada

**Correspondence to**
Dr Tatyana Mollayeva;
tatyana.mollayeva@utoronto.ca

## ABSTRACT

**Introduction** Reports on the association between comorbidity and functional status and risk of death in patients with traumatic brain injury (TBI) have been inconsistent; it is currently unknown which additional clinical entities (comorbidities) have an adverse influence on the evolution of outcomes across the lifespan of men and women with TBI. The current protocol outlines a strategy for a systematic review of the current evidence examining the impact of comorbidity on functional status and early-term and late-term mortality, taking into account known risk factors of these adverse outcomes (ie, demographic (age and sex) and injury-related characteristics).

**Methods and analysis** A comprehensive search strategy for TBI prognosis, functional (cognitive and physical) status and mortality studies has been developed in collaboration with a medical information specialist of the large rehabilitation teaching hospital. All peer-reviewed English language studies with longitudinal design in adults with TBI of any severity, published from May 1997 to April 2017, found through Medline, Central, Embase, Scopus, PsycINFO and bibliographies of identified articles, will be considered eligible. Study quality will be assessed using published guidelines.

**Ethics and dissemination** The authors will publish findings from this review in a peer-reviewed scientific journal(s) and present the results at national and international conferences. This work aims to understand how comorbidity may contribute to adverse outcomes in TBI, to inform risk stratification of patients and guide the management of brain injury acutely and at the chronic stages postinjury on a population level.

**PROSPERO registration number** CRD42017070033.

## Strengths and limitations of this study

► The study of comorbidity in traumatic brain injury (TBI) is important, as any additional clinical entity can change future life course and injury outcomes.

► To date, there has been no systematic review on the topic of comorbidity in TBI as it relates to all-cause mortality and functional status postinjury; the protocol outlines a strategy for a study that intends to fill the gap.

► Attention to known risk factors of adverse outcomes such as sex, age and TBI severity will permit advanced risk stratification and inform future prognostic studies.

► Biases associated with unequal sex and age distribution, residual confounding due to TBI-related or comorbidity-related treatment effect could not be avoided.

► Systematising prognostic data on comorbidity in TBI is essential for patients, healthcare providers, policy-makers and health researchers.

by the Public Health Agency projected that the indirect economic cost of a TBI due to working-age death and disability will increase from \$C7.3 billion in 2011 to \$C8.2 billion by 2031, far exceeding that of other common neurological conditions (eg, epilepsy, multiple sclerosis, Alzheimer's disease, combined with estimated \$C4.8 billion in 2011 and \$C5.8 billion in 2031, respectively).[4] TBI may also exacerbate pre-existing disorders, or expedite the development of, additional clinical conditions in both the older and younger populations, increasing direct and indirect costs associated with TBI.[5]

Of particular importance is that the presence of a comorbidity (ie, additional disease or illness coexisting with an index disease[6]) or multiple comorbidities in patients with TBI is common,[7–9] and is associated with high rates of hospitalisations, decreased functional status and all-cause mortality.[7–9] Studies have

## INTRODUCTION

Traumatic brain injury (TBI), defined as 'an alteration in brain function or other evidence of brain pathology, caused by an external force,'[1] is a major global health concern. According to WHO, death and disability from TBI are rising rapidly.[2] In the USA, the total annual cost of TBI was estimated to be US \$60.43 billion.[3] In Canada, a recent report

also shown that comorbid disorders may alter the treatment course of patients with TBI by affecting the treatments that these patients receive in both the acute and rehabilitation setting.[7–9] Specifically, among patients with a comorbid health condition, treating or managing comorbidities are often prioritised over addressing the TBI itself.[10–12] Likewise, the presence of any chronic comorbidity in TBI may lead patients to consume a disproportionate amount of healthcare resources.[13 14] While previous research has documented number of comorbidities in patients with TBI,[12 15 16] it is currently unknown which comorbid disorders pre-exist in TBI across ages, develop over time and which best predict outcomes related to functional (cognitive and physical) status and early and late postinjury mortality. It is also unknown if the presence of comorbidity changes the effect of traditional TBI risk factors for these outcomes, such as TBI severity and mechanism.[9 17–19]

Finally, it must be highlighted that a patient's sex and age has also been shown to drive differences in early mortality and functional recovery.[20 21] Previous research highlighted that case fatality ratios are elevated in older patients (60+ years of age) compared with younger patients, with sex differences observed in individuals 20–39 years of age.[21] Women are also at greater risk for the development of somatic and psychiatric comorbidity and associated functional decline postinjury.[22] Overall, despite the strong evidence that comorbidities lead to adverse outcomes and complications among patients with TBI, a data synthesis on this topic taking sex and age into account does not exist to date. This highlights the need to explore how age and sex associate with comorbidity risk factors in patients with TBI.

As such, this protocol is for a systematic review on the topic of comorbidity in adult patients with TBI that aims to: (1) examine the relationship between comorbid disorder(s) and change in function after TBI and death; (2) determine the prognostic value of clinical characteristics of patients with TBI at baseline on the development of short-term, intermediate-term and long-term adverse or beneficial outcome(s) and (3) review effects of comorbidity in the light of currently known risk factors of adverse outcomes (ie, sex, age and injury severity).

## METHODS AND ANALYSIS

The systematic review that this protocol describes will be conducted and reported in compliance with the Preferred Reporting Items for Systematic Reviews and Meta-Analyses guidelines.[23] In accordance with these guidelines, this systematic review protocol was registered with the International Prospective Register of Systematic Reviews (PROSPERO) on 22 June 2017.[24]

### Data sources and searches

In collaboration with TBI and rehabilitation experts, and a Medical Information Specialist, a comprehensive search strategy for prognostic studies of TBI outcomes

(ie, functional status and mortality) was developed (table 1). All English language peer-reviewed studies published between March 1997 and April 2017, with prospective or retrospective data collection and a longitudinal design, found through Medline, Central, Embase, Scopus, PsycINFO and bibliographies of identified articles will be included. Reference lists of included studies will be reviewed to identify any additional relevant studies. Search terms for each database are presented in online supplementary table 1.

### Comorbidity definition

Feinstein defined comorbidity as 'the existence or occurrence of any distinct additional entity during the clinical course of a patient who has the index disease under study'.[5] More recently, Valderas *et al* defined it as 'any additional condition that may occur during the clinical course of a patient who has an index condition that is the focus of interest'.[25] Canadian coding standards, the International Classification of Diseases, Version 10 (ICD-10-CA) defines comorbidity as 'a condition that coexists in addition to the most responsible diagnosis (MRDx) at the time of admission or that develops subsequently and meets at least one of the three criteria for significance: (1) requires treatment beyond maintenance of the pre-existing condition; (2) increases the length of stay by at least 24 hours and/or (3) significantly affects the treatment received'.[26] Currently, there is no gold standard for assessing comorbidity in patients with TBI and reports of comorbidity vary widely in the published work.[9 17–19] In population-based studies, comorbid conditions can be identified according to the ICD diagnostic codes or converted into summary comorbidity measures focused on selected conditions, such as the Charlson Comorbidity Index, Aggregated Diagnosis Groups, Elixhauser Comorbidity Index or the Total Illness Burden Index.[9 17–19] However, many papers focus on a single count of previously diagnosed chronic diseases that have shown a significant relation with death or functional outcomes in TBI population.[12–15] Given that there is no consensus on the most appropriate method to construct comorbidity in TBI or whether one type is preferred to another, this systematic review will set restrictions only towards comorbid disorders being diagnosed, excluding self-report. To account for inconsistencies between studies that will, at least partially, drive our results, close attention will be paid to definitions of comorbidity and assessment tools in each individual study when analysing and reporting results. In addition, the limiter zero time (baseline assessment) will be set at 6 months. This historical time limiter will be set to allow us to, indirectly, distinguish disorders that are chronic in nature (according to WHO, chronic disorders are those that require care beyond 6 months, such as diabetes and cardiovascular disorders), from those that may co-occur with TBI (neck injury, fractures, etc) and those that develop as a result of a TBI or associated impairments, both physical and psychological, such as anxiety and/or mood disorders, and infection

**Table 1** Prognosis, traumatic brain injury (TBI) and outcomes-related search terms (from Medline search strategy)

| | |
|---|---|
| Prognosis terms | Exp cohort studies/or exp prognosis/or exp morbidity/or exp mortality/or exp models, statistical/or prognos*.tw./or predict*.tw./or course*.tw./or diagnosed.tw/or death.tw./or cohort*.tw./or exp treatment outcome/or 'early termination of clinical trials'/or treatment failure/incidence/ |
| TBI terms | Exp brain injuries/or Craniocerebral Trauma/or exp Head Injuries, Closed/or exp Skull Fractures/or mTBI*2.tw,kw. or tbi*2.tw,kw. or concuss*.tw,kw. or ((head* or cerebr* or crani* or skull* or intracran*) adj2 (injur* or trauma* or damag* or wound* or swell* or oedema* or fracture* or contusion* or pressur*)).tw,kw. or ((brain* or cerebr* or intracerebr* or crani* or intracran* or head* or subdural* or epidural* or extradural*) adj (haematoma* or hematoma* or haemorrhag* or hemorrhag* or bleed*)).tw,kw. |
| Comorbidity terms | Exp Comorbidity/or exp Risk Adjustment/or (comorbid* or co morbid* or co morbid* or multimorbid* or multi morbid*).tw,kw. or (polypatholog* or poly-pathology*).tw,kw. or ((clinical* or medical*) adj3 complex*).tw,kw. or ((coexist* or co exist* or cooccur* or co-occur* or multipl*) adj3 (illness* or disease* or disorder* or condition* or complication* or diagnos* or risk*)).tw,kw. or (multidisease? or multi-disease? or (multiple adj (ill* or disease? or condition? or syndrom* or disorder?))).tw,kw. or ((several* or various or (two adj2 more) or concomitant or conjoined or concurrent) adj3 (morbid* or ill* or disease* or sick* or condition*)).tw,kw. Or "comorbidity-polypharmacy score".tw,kw. or ('charlson comorbidity index' or 'CCI' or 'CMI' or elixhauser or 'BOD index' or 'cumulative index rating scale' or 'CIRS' or 'Coroni-Huntley index' or 'DUSOI index' or 'Hallstrom index' or 'Hurwitz index' or 'Incalzi index', 'Kaplan index', 'Liu index', 'Shwartz index' or 'comorbidity-polypharmacy score').tw,kw. |
| Outcomes terms | Exp Mortality/or exp morbidity/or (morbidit* or mortalit*).tw,kw. or function*.mp. |

disorders. All attempts will be made to present results of acute comorbidity and chronic comorbidity associations with studied outcomes separately.

## Predictors and outcomes

Predictors will be collated into three domains: socio-demographic characteristics, TBI-related characteristics, and comorbidity. All comorbidity-related variables (comorbidity index severity or presence of comorbidity) will be treated as primary predictors; hypothesised socio-demographic (age and sex) and TBI-related variables (injury severity, mechanism and time since injury), if reported, will be considered as secondary predictors of our outcome(s) of interest. We will report on only those predictors that have been shown to be statistically significantly associated with our outcome(s) in at least one study and with reported quantitative data (ie, event rates, risk ratios (RRs), ORs or HRs) to measure the association between predictors and outcomes.

The following outcomes (either objectively documented or self-reported) will be considered: cognitive and physical status as assessed by the Glasgow Outcome Scale with or without extended scores, Disability Rating Scale, Functional Independence Measure, the Functional Status Examination or any other standardised functional measurement and mortality. In order for the previous objective (ie, functional status) to be considered, the study has to define at least two time points' scores of functioning and/or has to provide details on when (at which time point since injury) a loss/gain/plateau of function has been defined as a decline/improvement/stability. The definition of functional status should vary among studies, and the reported minimum change, expressed in percentage change, will be abstracted. For example, if the functional measure has a score in a range from 0 to 10, and the minimum change reported by researchers for the sample was 1, we will assign a 10% 'change' on this group's functional capabilities.

## Inclusion and exclusion criteria

We will include studies that (1) primarily study comorbidity as it relates to our outcome(s) of interest and (2) targeted adult patients (the mean age minus SD is ≥18 years of age) with a diagnosis of TBI on the basis of the accepted definitions (not self-report) and followed them for any period of time. Studies of brain injury of only traumatic origin will be considered. Studies will neither be excluded based on the setting in which the research took place (acute care, rehabilitation setting, community, etc) nor means of diagnosis of comorbidity. However, the following studies will be excluded: (1) more than 50% of participants had pre-existing TBIs or severe comorbidity (ie, neurological or psychiatric diseases) at the baseline assessment, and if the subgroup with incident comorbidity and the patient outcome data could not be extracted independent of pre-existing cases (ie, present before TBI) and (2) study designs/formats in letters to editors, reviews without data, case reports or public

reports, conference abstracts articles with no primary data, studies that focus on therapeutic interventions and theses.

### Zero time (baseline assessment)

The nature of our objectives related to development of adverse outcomes in the TBI population (ie, prognostic factors), raises the issue of zero-time bias. In prognostic studies, testing should start at a defined point, called zero time. Designated zero times (ie, baseline or first assessment) vary between studies,[26] where the majority of research from acute care/emergency studies performed baseline assessment within the first month after injury and majority of rehabilitation or community studies of prognosis performed baseline assessment prior to or at 6 months postinjury mark.[27] The limiter to zero time has been set at 6 months.

### Study selection

Two independent researchers (CX, SH) will assess study titles and abstracts. If the title or abstract suggests that the study might meet the inclusion criteria, both reviewers will assess the full article. Differences of opinion will be resolved by group discussion (CX, SH and TM), with the goal to reach consensus in each case. Studies failing to meet the inclusion criteria will be excluded and the reason will be reported.

### Data extraction and quality assessment

Study quality will be assessed independently by two researchers (TM, CX) using guidelines for assessing prognostic studies.[28] First, two researchers will independently assess the items related to potential sources of bias, namely: (1) study participation and attrition; (2) prognostic factor and outcome measurements; (3) confounding measurement and account and (4) analyses. Then, the same two reviewers will judge the presence of potential biases as 'Yes', 'Partly', 'No' or 'Unsure'. Following these steps, the Scottish Intercollegiate Guidelines Network[29] methodology will be implemented where '++' will be assigned to each study when all or most of the quality criteria were fulfilled (allowing one 'Partly' while appraising all potential sources of bias); '+' when some of the criteria were fulfilled and '−' when few or no criteria fulfilled (at least one 'Yes'). In our review, we will refer to group '++' as 'high quality studies' and group '+' as 'moderate quality studies'. We will abstract data on the relationships between our outcomes of interest and primary (ie, comorbidity) and secondary (ie, sociodemographic, TBI-related) predictors only from studies with sufficient quality (ie, 'high' and 'moderate' quality studies).[29]

### Dealing with missing data

Primary authors will be contacted in cases of missing data. The proportion of missing data will be reported along with reasons where indicated. In the case of duplicate publications and companion papers of a primary study, we will attempt to yield maximum scientific information by abstraction of all available data. However, original publication (usually the earliest publication version) will take priority in data analysis.

### Dealing with publication bias

The predisposition of journals to favour publication of positive reports over negative investigative findings and the reticence of authors to publish poor outcomes may lead to a high chance that results of studies included in this review may be affected by these publication biases. The most commonly used method to assess potential publication bias is the construction of a funnel plot, which is not an optimal methodology in highly heterogeneous studies that precludes expecting a symmetrical funnel shape.[30] To account for potential publication bias, we will apply an expert opinion methodology[31] to inform the study selection process. We will ask researchers and clinicians from our team with expertise in TBI about the probability of publication for small and large sample size studies that considered the effect (ie, positive or negative) of any comorbidity in relationship to outcomes, and will report the average in their response. We will then apply the selection model on the published studies and calculate an estimate from the published studies (without making any adjustments for publication bias). We will also report on the quality of the study and funding information. Such an approach is not confounded by heterogeneity as in the case of the funnel-plot approach.

### Data synthesis and analysis

For the unadjusted analysis (step 1), we will extract data from all studies that reported the effect of comorbidity on mortality and functional status and report number of events (death and functional decline/gain/plateau) relative to the total number of participants in the group with comorbid disorder(s) and control groups. The time frame of follow-up assessments will be categorised into (1) short term, that is, up to 3 months postinjury mark; (2) intermediate, that is, 3–12 months (ie, up to 1-year inclusive) and (3) long term, that is, >1-year postinjury. These points were arbitrarily set. We reserve the right to adjust follow-up time frames based on time stratification applied in the included studies.

This stage will be followed by adjusted analysis (step 2), where we will extract and analyse quantitative data (ie, ORs, RRs and HRs) that will be adjusted for hypothesised key confounders (age, sex, TBI mechanism and severity and/or baseline functional status) reflecting the association between comorbidity and our outcomes of interest. Some of these variables, such as age, sex and TBI severity, are considered as key confounders, and will be included in our list of required adjusted variables for a study to be included in our primary analysis.[28] If the key variables are not included in the final adjusted model, control for confounding variables will be determined to be inadequate, which will be reflected in a risk of bias assessment table.[28] Where possible, we will perform a meta-analytic analysis: pooled-effect outcomes for each

group of comorbid disorders will be calculated using inverse variance methods with random effects models[29] and expressed as ORs and 95% CIs. Heterogeneity will be assessed using the $I^2$ statistic. p Values of 0.05 or less will be considered as statistically significant. In the case of statistical or clinical diversity in definitions of comorbidity and/or TBI, population of interest and the statistical methodology used to quantify association in the studies, meta-analysis will not be performed and we will use a best evidence synthesis approach, synthesising findings from studies with sufficient quality through tabulation and qualitative description.[32]

## DISCUSSION
### Ethics and dissemination

This systematic review aims to investigate the relationship between comorbidity and functional outcomes and death within short, moderate and long-term time frames after TBI, as well as determining the prognostic value of comorbidity, sociodemographic (ie, age, sex) and injury-related characteristics on these outcomes. The strength of this systematic review and research programme is in its methodology, making it possible to identify associations longitudinally, thus improving the quality of inductive inferences regarding the natural progression of associative values of hypothesised predictors on outcomes in patients with TBI of various severities. Furthermore, our protocol was registered in PROSPERO and was designed in keeping with best practice methods where the multilevel risk of bias assessment[24] will allow us to detect the main flaws in the individual studies' design and inform on the future research of comorbidities in TBI. Moreover, the clinical criteria for the diagnoses of TBI and comorbidity will be collected and reported, as it is expected that they have a significant impact on the study results. Finally, multilevel knowledge translation activities throughout this research activity will be performed, ensuring that these results reach their intended knowledge users.

### Limitations

The present study includes the following limitations: (1) the limiter of 6 months for zero time may not be optimal as some comorbid disorder (demyelinating, degenerative, etc) may take longer time to develop; (2) the assumption of expected heterogeneity in the primary studies with respect to TBI-related characteristics (ie, injury/localisation of injury, time since injury) with severe TBI cases expect to be under-represented; likewise severe comorbid disorders may precluded patients with TBI with milder severities of injury to participate in research, limiting the precision of estimates of risk for severe TBI and comorbidity cases; (3) potentially unequal sex and age distribution in primary studies, given that historically TBI has been considered an injury of younger men and older women; (4) residual confounding due to TBI-related or comorbidity-related treatment effect; (5) excepted complexity of assessing the risk of short-term functional change (ie, fluctuation)

and (6) additional limitations relate to the exclusion of grey literature, articles published in languages other than English and limiting our searches to the past 20 years; this decision was based on the extensive number of studies identified within the databases searched, changes applied to clinical classifications and definitions of TBI, as well as limited empirical evidence about the potential impact of selective searching and inclusion of earlier works on the results of systematic reviews.[33]

Despite these limitations, this protocol is for a review, that is, the first that comprehensively synthesises evidence on prognostic value of comorbidity in patients with TBI, aiming to enrich science and advance care provided to patients with comorbid disorders stemming from TBIs.

### Implications

The number of people surviving TBIs is increasing. While the neurological consequences of TBI are well described, evidence is emerging on associations between brain injury, comorbid disorders and adverse short-term and long-term outcomes postinjury.[34 35] The significant economic and human costs of TBIs merit the call for systematic efforts to understand all factors that contribute to adverse postinjury outcomes, including comorbidity[36]; all with the goal to allow better risk stratification to guide management of brain injury acutely and at the chronic stages postinjury on a population level.

**Acknowledgements** We gratefully acknowledge the involvement of Jessica Babineau, Information Specialist at the Toronto Rehabilitation Institute- University Health Network for her comprehensive literature search.

**Contributors** Protocol concept and design: TM, AC. Registry PROSPERO: CX. Acquisition of data: TM. Administrative, technical and material support: TM, CX, SH. Statistical analysis approach: TM, VC, ZJH, MS, ME. Drafting of the manuscript: TM. Critical revision of the manuscript for important intellectual content: All authors.

**Funding** The study was funded by the National Institutes of Health (NIH) under Award 1R21 HD08106-01 (principal investigator: Dr Angela Colantonio). AC is supported by the Canadian Institutes for Health Research Grant–Institute for Gender and Health (grant no CGW-126580). TM is supported by the postdoctoral research grant Alzheimer's Association (AARF-16-442937).

**Competing interests** None declared.

**Provenance and peer review** Not commissioned; externally peer reviewed.

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
