## [Reviewer comments · BMJ Open]

ARTICLE DETAILS

TITLE (PROVISIONAL)	Comorbidity and outcomes in traumatic brain injury: Protocol for a systematic review on functional status and risk of death.
AUTHORS	Mollayeva , Tatyana ; Xiong, Chen; Hanafy, Sara; Chan, Vincy; Hu, Zheng Jing; Sutton, Mitchell; Escobar, Michael; Colantonio, Angela

VERSION 1 – REVIEW

REVIEWER	Meshkini, Mohammad Road Traffic Injury Research Center, Tabriz University of Medical Sciences, Tabriz, Iran There is no competing of interest to declare with this review and protocol.
REVIEW RETURNED	22-Jul-2017

GENERAL COMMENTS	I really appreciate your concerns about TBI subject, and Hope your work would enlighten the path for physicians' decision making and further future researches. Thanks a lot for your patience and invaluable systematic review work; I hope to read the paper ASAP.
---

REVIEWER	David Williamson Hopital du Sacre-Coeur de Montreal, Canada
REVIEW RETURNED	09-Aug-2017

GENERAL COMMENTS	This is a well written and rigorous protocol. I look forward to reading the results. General comments: Co-morbid disorders can exist before a patient suffers a TBI (i.e. COPD, Heart failure, Cognitive dysfunction), develop along with TBI (i.e. SCI, other types of injuries) or as a complication (i.e. neuropsychiatric complications, chronic pain, etc,) or develop over the long-term following TBI. It remains unclear to me how you will differentiate the association between the acquisition of these comorbidities in patients with TBI and outcomes. Minor comments: p.10 Line 18 - What is the justification for limiting the search strategy to 1997-2017? p.10 Line 22 - Will you be searching any grey literature or open access database such as DOAJ? Please state
---

	p.10 Line 15 - Although including only English language studies is simpler, I fear this might induce selection bias. Please justify the exclusion or include other languages. p.11 Line 30 - Will you report comorbidities with association in both univariate and multivariate analysis? P.12 Line 20 - What will you do about studies which included mixed population of brain diseases that also include TBI. Will you exclude them, accept them if 50% are TBI? Please specify P.13 Line 23- Please look-out for information bias regarding studies using ICD codes. P.13 Line 25- How will you handle any disagreements between the 2 reviewers? P.14 Line 53- Are any sensitivity analyses planned to assess the robustness of the findings? P.15 Line 6 - What will you do in the presence of significant statistical (>50%), methodological or clinical heterogeneity? Will you still pool the data? P.15 Line 10 - Is there any plan to assess publication bias in any way?
--	---

VERSION 1 – AUTHOR RESPONSE

Reviewer: 1

Reviewer Name: Meshkini, Mohammad

Institution and Country: Road Traffic Injury Research Center, Tabriz University of Medical Sciences, Tabriz, Iran

Competing Interests: There is no competing of interest to declare with this review and protocol.

Dear Authors,

I really appreciate your concerns about TBI subject, and Hope your work would enlighten the path for physicians' decision making and further future researches.

Thanks a lot for your patience and invaluable systematic review work; I hope to read the paper ASAP.

Response: We thank the reviewer for taking the time to review the protocol and his positive feedback.

Reviewer: 2

Reviewer Name: David Williamson

Institution and Country: Hopital du Sacre-Coeur de Montreal, Canada

Competing Interests: None declared

This is a well written and rigorous protocol. I look forward to reading the results.

General comments:

Co-morbid disorders can exist before a patient suffers a TBI (i.e. COPD, Heart failure, Cognitive dysfunction), develop along with TBI (i.e. SCI, other types of injuries) or as a complication (i.e. neuropsychiatric complications, chronic pain, etc.) or develop over the long-term following TBI. It remains unclear to me how you will differentiate the association between the acquisition of these comorbidities in patients with TBI and outcomes.

Response: We thank the reviewer for taking the time to review the protocol and his comments. We agree with Dr. Williamson that additional clinical entities (comorbid disorders) may be present before the injury, co-occur with injury (additional injuries) or subsequently develop after the injury, as the person ages and more comorbid condition accumulate. This is represented in the definitions of important contributors that work on the topic in the past. Feinstein, a father of comorbidity, defined it in 1970 as “any distinct additional clinical entity that has existed or that may occur during the clinical course of a patient who has the index disease under study”[1]. Valderas in 2009 defined comorbidity as “any additional condition that may occur during the clinical course of a patient who has an index condition that is the focus of interest” [2]. Finally, Canadian coding standards, the International Classification of Disease Version 10 (ICD-10-CA), defines comorbidity as “a condition that coexists in addition to the most responsible diagnosis (MRDx) at the time of admission or that develops subsequently and meets at least one of the three criteria for significance”: (1) requires treatment beyond maintenance of the pre-existing condition; (2) increases the length of stay (LOC) by at least 24 hours; and/or (3) significantly affects the treatment received” [3]. To account for these discrepancies that will, at least partially, drive our results, we will closely examine definitions of comorbidity in each individual research when analyzing the data, and report on definition in our main table. In addition, the limiter zero-time (baseline assessment) has been set at six-month. This historical time limiter allow us to indirectly distinguish disorders that are chronic in nature (according to the World Health Organization, chronic disorders are those that require care beyond six months, such as diabetes, cardiovascular disorders, etc.), from those that may co-occur with TBI (neck injury, fractures, etc.) and those that develop as a result of a TBI or associated impairments, both physical and psychological, such as anxiety and/or mood disorders, infection disorders, etc.. All attempts will be made to present results of acute comorbidity and chronic comorbidity associations with studied outcomes separately.

References:

1 Feinstein AR. The pre-therapeutic classification of co-morbidity in chronic disease. *J Chron Disease* 1970; 23: 455–468.

2 Valderas JM, Starfiels B, Sibbald B, Salisbury C, Roland M. Defining comorbidity: implications for understanding health and health services. *Ann Fam Med* 2009; 7:357-363

3 Canadian Coding Standards for Version 2015 ICD-10-CA and CCI, assessed June 25, 2017 at: https://secure.cihi.ca/free_products/Coding%20standard_EN_web.pdf

Minor comments:

p.10 Line 18 - What is the justification for limiting the search strategy to 1997-2017?

Response: We thank the reviewer for raising the issues for limiting the search strategy to 1997-2017. The rationale for this was mainly due to the significant number of abstracts identified within our search databases even after removal of duplicates, i.e. more than 9,000 – these searches are by far the most comprehensive of all done on any topic in TBI. In addition, there were changes in the middle 1990th to clinical definitions of TBI, which is expected to increase the heterogeneity of results and would impact the ability to perform meaningful data synthesis. Finally, limited empirical evidence exists about the potential impact of selective searching and inclusion of earlier works on the results of systematic reviews.

We have added the following paragraph to the limitations section:

Additional limitations relate to the exclusion of grey literature, articles published in languages other than English, and limiting our searches to the past 20 years; this decision was based on the extensive number of studies identified within the databases searched, changes applied to clinical classifications and definitions of TBI, as well as limited empiric evidence about the potential impact of selective searching and inclusion of earlier works on the results of systematic reviews.

p.10 Line 22 - Will you be searching any grey literature or open access database such as DOAJ?
Please state

Response: We thank the reviewer for raising the issue of grey literature and open databases. We are aware that the methodological standards for systematic reviews recommend extensive searching to address the potential for publication bias and to produce accurate and valid estimates of effect; this has also been reflected in the recent Cochrane guidelines. The rationale for excluding grey literature, and other sources of information in our study was based on two reasons. First, the significant number of abstracts identified within our search databases even after removal of duplicates, (i.e. more than 9,000) –are by far the most comprehensive of all searches done on any topic in TBI; in comparison, the number of identified works in the initial screen generally does not exceed 5,000 abstracts. Second, a recent systematic review on the topic has provided empirical evidence on the limited value of searching for and including grey literature (i.e., studies published in languages other than English, unpublished studies and dissertations). Inclusion of these study types may have an impact in situations where there are few relevant studies, or where there are questionable vested interests in the published literature. These issues are not concerns without research question.

Reference: Hartling L, Featherstone R, Nuspl M, Shave K, Dryden DM, Vandermeer B. Grey literature in systematic reviews: a cross-sectional study of the contribution of non-English reports, unpublished studies and dissertations to the results of meta-analyses in child-relevant reviews. Reference: BMC Med Res Methodol. 2017 Apr 19;17(1):64

We have added the following paragraph to the limitations section:

Additional limitations relate to the exclusion of grey literature, articles published in languages other than English, and limiting our searches to the past 20 years; the decision was based on the extensive number of studies identified within the databases searched, changes applied to clinical classifications and definitions of TBI, as well as limited empirical evidence about the potential impact of selective searching and inclusion of earlier works on the results of systematic reviews.

p.10 Line 15 - Although including only English language studies is simpler, I fear this might induce selection bias. Please justify the exclusion or include other languages.

Response: We thank the reviewer for raising the issues for limiting the search strategy to English studies only. The rationale for this was mainly due to lack of funding to employ researchers with languages other than English and train them in methodology to perform a systematic review. This limitation has been highlighted in the limitations section.

p.11 Line 30 - Will you report comorbidities with association in both univariate and multivariate analysis?

Response: We will emphasize and report associations in multivariate analysis, where variables, such as age, sex, and TBI severity are considered as key confounders. If the key variables are not included in the adjusted model, control for confounding variables will be determined to be inadequate, and this will be reflected in a risk of bias assessment table.

P.12 Line 20 - What will you do about studies which included mixed population of brain diseases that also include TBI. Will you exclude them, accept them if 50% are TBI? Please specify

Response: Studies that reported on acquired brain injury and/or other brain disorders without reporting results on TBI separately will be excluded. This is because disorders that are of traumatic origin differ in natural history, prognosis, and risk factors than those injuries of a non-traumatic origin. We will consider studies that present results for TBI group separately from non-TBI.

We have added the following sentence to the Inclusion and exclusion criteria section:
Studies of brain injury of only traumatic origin will be considered.

P.13 Line 23- Please look-out for information bias regarding studies using ICD codes.

Response: We thank the reviewer for raising the issue of information bias using ICD codes. We are aware on this, and will consciously report on ICD codes used to define TBI and comorbidity. Both will be reported in a main table.

P.13 Line 25- How will you handle any disagreements between the 2 reviewers?

Response: Differences in opinion will be resolved by group discussion, with the goal to reach consensus in each case. This is stated in the protocol.

P.14 Line 53- Are any sensitivity analyses planned to assess the robustness of the findings?

Response: Sensitivity analysis in its true meaning, as a part of a meta-analysis, will not be performed due to expected heterogeneity present at numerous levels. This includes the heterogeneous study methodology (i.e. cohort, case-control, RCT, case series), population of interest in terms of injury severity, etc., timing of assessment, as well as type of comorbidity and utilised definitions of comorbidity and TBI. This heterogeneity at the multiple levels precludes us from performing a meaningful meta-analysis and a formal sensitivity analysis. Nonetheless, the robustness of the observed results to all the assumptions specified in this protocol will be ensured by presenting the combined and stratified results by injury severity, sex and age, to determine whether data stratification has an effect on the relationship between comorbidity and outcomes of interest. We will also take into account study quality, and repeat analysis by reporting lower quality studies separately from high quality studies.

P.15 Line 6 - What will you do in the presence of significant statistical (>50%), methodological or clinical heterogeneity? Will you still pool the data?

Response: Thank you for raising the issue of study heterogeneity. We will attempt to perform all meaningful combinations/stratification of results where possible in an attempt to reduce heterogeneity and investigate effect of comorbidity within similar injury severity and age, as it relate to our outcomes of interest. If not possible, i.e., in the presence of significant heterogeneity (statistical or clinical), meta-analysis will not be performed, and we will utilise best evidence to report our results.

We have added the following sentence and the citation to the manuscript:

In the case of significant diversity in definitions of comorbidity and/or TBI, population of interest, and the statistical methodology used to quantify association in the studies, meta-analysis will not be performed and we will use a best-evidence synthesis approach, synthesizing findings from studies with sufficient quality through tabulation and qualitative description. 33

33. Slavin RE. (1995) Best evidence synthesis: an intelligent alternative to meta-analysis. *J Clin Epidemiol*, 48: 9–18.

P.15 Line 10 - Is there any to plan to assess publication bias in any way?

Response: Thank you for raising the issue of a potential publication bias. We are aware of the predisposition of journals to favour publication of positive reports over negative investigative findings,

and the reticence of authors to publish poor outcomes may lead to a high chance that results of studies included in this review may be affected by these publication biases. The most commonly used method to assess potential publication bias, i.e., the construction of a funnel plot, is not an optimal methodology. The heterogeneity from study to study because of differences in study protocol, study quality, injury severity, and patient characteristic precludes us to expect a symmetrical funnel shape. To account for potential publication bias, we will apply the selection model approach, i.e., an expert opinion methodology, to inform the study selection process. We will ask researchers and clinicians with expertise in TBI from our team about the probability of publication for small and large sample size studies that considered effect of any comorbidity (positive or negative) in relationship to clinically relevant outcomes, and will report the average in their response. We will then apply the selection model on the published studies and calculate an estimate from the published studies (without making any adjustments for publication bias). We will also report on the quality of the study and funding information. Such an approach is not confounded by heterogeneity as in the case of the funnel-plot approach.

The following section with references have been added to the protocol:

Dealing with publication bias

The predisposition of journals to favour publication of positive reports over negative investigative findings and the reticence of authors to publish poor outcomes may lead to a high chance that results of studies included in this review may be affected by these publication biases. The most commonly used method to assess potential publication bias is the construction of a funnel plot, which is not an optimal methodology in highly heterogeneous studies that precludes expecting a symmetrical funnel shape.³² To account for potential publication bias, we will apply an expert opinion methodology,³³ to inform the study selection process. We will ask researchers and clinicians from our team with expertise in TBI about the probability of publication for small and large sample size studies that considered effect (i.e., positive or negative) of any comorbidity in relationship to outcomes, and will report the average in their response. We will then apply the selection model on the published studies and calculate an estimate from the published studies (without making any adjustments for publication bias). We will also report on the quality of the study and funding information. Such an approach is not confounded by heterogeneity as in the case of the funnel-plot approach.

References:

1. Terrin N, Schmid CH, Lau J. In an empirical evaluation of the funnel plot, researchers could not visually identify publication bias. *J Clin Epidemiol.* 2005; 58(9):894-901.
2. Mavridis D, Salanti G. Exploring and accounting for publication bias in mental health: a brief overview of methods. *Evid Based Ment Health.* 2014; 17(1):11-5.